# The Bellinge data set: open data and models for community-wide urban drainage systems research

Agnethe Nedergaard Pedersen[1,2], Jonas Wied Pedersen[2], Antonio Vigueras-Rodriguez[3], Annette Brink-Kjær[1], Morten Borup[2], Peter Steen Mikkelsen[2]

[1]VCS Denmark, Odense, 5000, Denmark
[2]DTU Environment, DTU, Kgs. Lyngby, 2800, Denmark
3Department of Civil Engineering, Universidad Politécnica de Cartagena, 30203 Cartagena, Spain

*Correspondence to*: Agnethe Nedergaard Pedersen (anp@vandcenter.dk)

**Abstract.** This paper describes a comprehensive and unique open-access data set for research within hydrological and hydraulic modelling of urban drainage systems. The data comes from a mainly combined urban drainage system covering a 1.7 km$^2$ area in the town of Bellinge, a suburb to the city of Odense, Denmark. The data set consists of up to 10 years of observations (2010–2020) from 13 level meters, one flow meter, one position-sensor and four power sensors in the system, along with rainfall data from three rain gauges and two weather radars (X- and C-band), and meteorological data from a nearby weather station. The system characteristics of the urban drainage system (information about manholes, pipes etc.) can be found in the data set along with characteristics of the surface area (contour lines etc.). Two detailed hydrodynamic, distributed urban drainage models of the system are provided in the software systems Mike Urban and EPA SWMM. The two simulation models generally show similar responses, but systematic differences are present since the models have not been calibrated. With this data set we provide a useful case that enables independent testing and replication of results from future scientific developments and innovation within urban hydrology and urban drainage system research. The data set can be downloaded from https://doi.org/10.11583/DTU.c.5029124, (Pedersen et al., 2021a).

## 1 Introduction

Scientific progress related to urban hydrology and urban drainage systems research is slowed down by a lack of open data within the field, and the need for open data and transparency is thus increasingly being emphasised (Moy de Vitry et al., 2019; Vonach et al., 2019). Urban drainage systems are essential for protecting the environment as well as human health and property. They typically represent the largest capital investments in infrastructure in cities and the most cost-efficient in terms of socio-economic gain (Hutton et al., 2007). Utility companies typically own the urban drainage systems and administer the right to extract and share asset data and sensor observation data from the systems. Sharing data with a larger community is a complex task as data requires metadata and local knowledge, and because data is often hidden in various difficult to access data systems in the utility. Utility companies have traditionally had little interest in making their data publicly available, and sometimes even reject this for security or publicity reasons. They are however usually interested in collaborating with local universities,

and therefore most published studies are based on case data that are not shared with the broader international scientific community. This makes it virtually impossible to compare the performance of different methods, which makes it difficult to reach a scientific consensus that can allow us to start focusing on future innovation and to initiate the capacity building worldwide that is needed to ensure better urban drainage asset management. In the broader hydrological community there have been several major efforts to provide "community-wide data" with the explicit purpose of improving research and innovation in their field, e.g., CAMELS (Addor et al., 2017) and MOPEX (Schaake et al., 2006). A large part of these impressive data sets consists of satellite data or derivatives from these, which have made it possible to obtain data on close to continental scale. This is not possible within the field of urban drainage, since the most important parts of the systems are hidden underground, and the data describing these are either non-existing or hidden in various utility companies' data systems. The basis for open data sets within urban drainage is thus bound to be the utility company.

Ideally, an open-access dataset for an urban drainage system should be able to stand alone without the need for direct contact with the utility company, so that any researcher gets the same level of access to information. This implies that the data set inevitably will contain more information than needed for any single study. The minimum requirement for such a data set is a detailed description of the drainage system given by asset documentation, such as dimensions and location of all pipes and structures, as well as time series of observed inflow to the system (rainfall observations and/or consumer's wastewater production). Within the past few years, several research groups have presented surrogate modelling studies on each their case areas and using their own specific case data, which makes it impossible to compare the methods directly (Kroll et al., 2017; Ledergerber et al., 2019; Mahmoodian et al., 2018; Thrysøe et al., 2019; Wolfs and Willems, 2017). Within the research field of Real Time Control (RTC) of urban drainage systems, the lack of open data sets has led to the development of synthetic test models, such as the Astlingen network (Schütze et al., 2017; Sun et al., 2020) and the Pystorms networks (Rimer et al., 2019). While such synthetic networks are useful due to their stringent focus on the most relevant processes for the purpose at hand, the usage of actual networks to benchmark the performance of RTC methods would help the end-users in the utility companies to decide which methods to implement for their specific system. A comprehensive data set of one structure, heavily monitored during 5 days of experiment has been released (Moy De Vitry et al., 2017), and similar is needed for networks systems. Time series of observations of levels and flows in the system will also be necessary for many investigations of e.g., model calibration techniques (Krebs et al., 2013; Vonach et al., 2019), development of improved skill scores (Bennett et al., 2013), uncertainty analysis (Deletic et al., 2012), techniques for data quality control (Kirstein et al., 2019; Therrien et al., 2020), development of data-driven models and machine learning (Carbajal et al., 2017; Eggimann et al., 2017; Palmitessa et al., 2021) and software sensors (Fencl et al., 2019). Other areas that can be inspired by open data sharing could be the construction of digital ecosystems (Sarni et al., 2019) and digital twins (Pedersen et al., 2021b; Therrien et al., 2020). The more complete and more diverse the data set is (spatially and temporally), the more research potential there will be.

The current article describes an open data set suitable for urban drainage research and education. The data set is described in accordance with the FAIR principles of open and documented data, which requires data to be Findable, Accessible, Interoperable and Reusable (Wilkinson, 2016) in order to fully support reproducible research in computational hydrology (Hutton et al., 2016; Stagge et al., 2019). The utility company VCS Denmark (referred to as "VCS" in the rest of the paper) provided most of the data with support of hydrological and meteorological data from the Danish Meteorological Institute (DMI) (DMI, 2020), rain gauge data from The Water Pollution Committee (WPC) of The Society of Danish Engineers (DMI and IDA, 2020) and geospatial data from the Danish Agency for Data Supply and Efficiency (DADSE) (DADSE, 2020). VCS has for many years focused on documenting its systems and procedures through detailed registration of assets and systematized collection of sensor observation data. The presented data set is from a 1.7 km$^2$ suburban area served by a combined urban drainage system with more than 10 years of observational data available. Orthophotos from 2010 to the present show no significant urban development in the areas that are connected to the combined sewer system. This gives a unique opportunity to use the same model for a 10-year period with little structural model uncertainty induced by changes of the urban layout. The specific properties of the sewer system (pipe diameters, basin volumes, actuator settings, etc.) are shared in the shape of two distributed urban drainage models, since these are more accessible for most potential users than an asset database. The two simulation models are constructed in the software programs Mike Urban (MU) (DHI, 2020) and Storm Water Management Model (SWMM) (EPA, 2020). The MU model is used by VCS in the daily planning and operations work of the utility company and the SWMM model of the same system has been constructed for this publication due to its free and open source nature. The selection of data and models provided here aims to be as "open-minded" as possible, and we believe it can be used to initiate research across a range of highly relevant topics, and also inspire discussions among water utilities on the benefits of high-quality data acquisition and modelling.

The paper is organized as follows: The case area and its urban drainage system is described in Sect. 2, while Sect. 3 describes the sensor observations and Sect. 4 the two distributed urban drainage models. Sect. 5 provides a brief comparison of the two models with data from selected locations for three example rain events and discusses possible ways of improvement, followed by a discussion of future potential research use of the data set (Sect. 6), an overview of the data repository (Sect. 7) and our conclusions (Sect. 8).

**2 System description**

The presented case area consists of the three suburban towns, Brændekilde, Bellinge and Dyrup, which are located in the south-western outskirts of Odense on the island of Fyn, Denmark (Figure 1A) at 55°24' northern latitude and 10°23' eastern longitude, in a temperate climate with seasons winter (December–February), spring (March–May), summer (June–August) and autumn (September–November). The municipality of Odense has approximately 200,000 inhabitants and an area of 30,400 ha of which about half is developed. VCS is the local utility company responsible for operating and managing most of the

water supply and all the urban drainage and wastewater infrastructure in the municipalities of Odense and Nordfyn, including the central Ejby Mølle Water Resource Recovery Facility (WRRF, also referred to as wastewater treatment plant), which collects combined and separate wastewater from a 112 km$^2$ service area, including Brændekilde-Bellinge-Dyrup in the most upstream part of the Ejby Mølle sewer catchment.

## 2.1 Characteristics of the area

Topographically, the area is flat with terrain dropping from 50 meters above sea level in the west to 20 m in the east. It is located in the downstream part of the Odense River catchment, in-between Odense River and its tributary Borreby Møllebæk, which receive combined sewer overflow from Bellinge and Brændekilde during heavy rain (Figure 1B,C). All sensors are located in Brændekilde and Bellinge (Figure 1D), the total surface area contributing with rainfall-runoff to the sewer system (upstream from sensor G71F06R in Bellinge) being 1.72 km$^2$ with paved surfaces covering 0.55 km$^2$. The case area has 1,800 households with approximately 4,000 inhabitants, which mostly live in detached single-family houses built from 1960 to 1980 and discharge a total average wastewater load of 501 m$^3$/day. Downstream from Bellinge, a main interceptor pipe runs along Odense River to the Ejby Mølle WRRF, as seen in Figure 1B. The small town Dyrup is included in the models to ensure realistic hydraulic conditions in the interceptor pipe. A total average of 570 m$^3$/day of wastewater and 1 km$^2$ surface area contributes to the boundary conditions downstream of G71F06R, herein stormwater from an area of 0.61 km$^2$ coming from Dyrup.

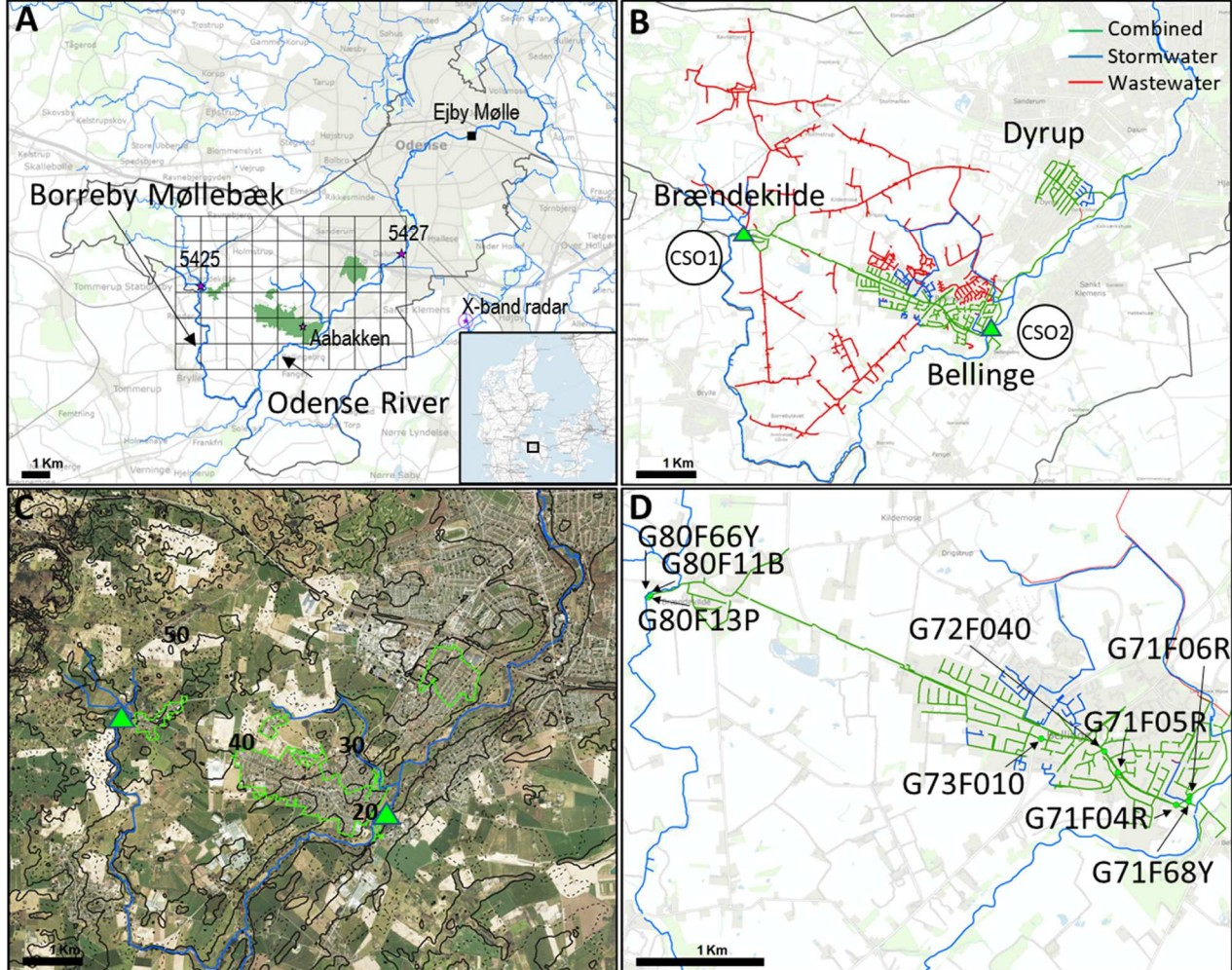

**Figure 1: Maps of the case area, indicating rivers as blue lines. A: Locations of Brændekilde, Bellinge and Dyrup (dark green areas) in the upstream part of the sewer catchment of Ejby Mølle WRRF (black outline), locations of rain gauges (stars) and locations of X-band radar grid cells (grid). B: The pipe network in the area, distinguishing combined sewers (green) from separate sewers for stormwater (blue) and wastewater (red). Combined sewer overflow locations (green triangles). C: Ortho-photo of the area along with contour lines, indicating combined sewer overflow locations (green triangles). D: Locations of the in-sewer sensors in Brændekilde and Bellinge together with the combined and stormwater system (wastewater pipes are hidden for simplicity). Background maps provided by DADSE (2020).**

## 2.2 Climate and meteorology of the area

The area has a warm, temperate climate. Several meteorological variables are available for free through DMI's open data platform (DMI, 2020). The nearest DMI weather station is at Årslev approximately 10 km east of the case area (outside the area shown on Figure 1). The historical data includes time series (10 min resolution) of temperature, relative humidity, wind speed, amount of time with direct sunlight, and solar radiation. The annual rainfall depth varied between 530 and 820 mm (Figure 2) over the past 10 years, with the predominant westerly wind direction bringing many large frontal systems over the

catchment, especially during autumn and winter. Winters tend to be mild with a relatively short snow season, while spring is slightly drier than the rest of the year. The daily-average temperature has been as low as -10°C during winter and up to +20°C during summer. High-intensity, convective rainfall events mostly occur during the summer months. The maximum recorded intensity in the 10 years data set from Årslev is 11.8 mm/10 minutes, corresponding to a mean of 1.18 mm/min or 19.7 μm/s during 10 minutes. Solar radiation, wind speed and humidity data are also provided from the Årslev station, thereby allowing evapotranspiration to be calculated. The wind speed and relative humidity is in general highest during the autumn and winters period.

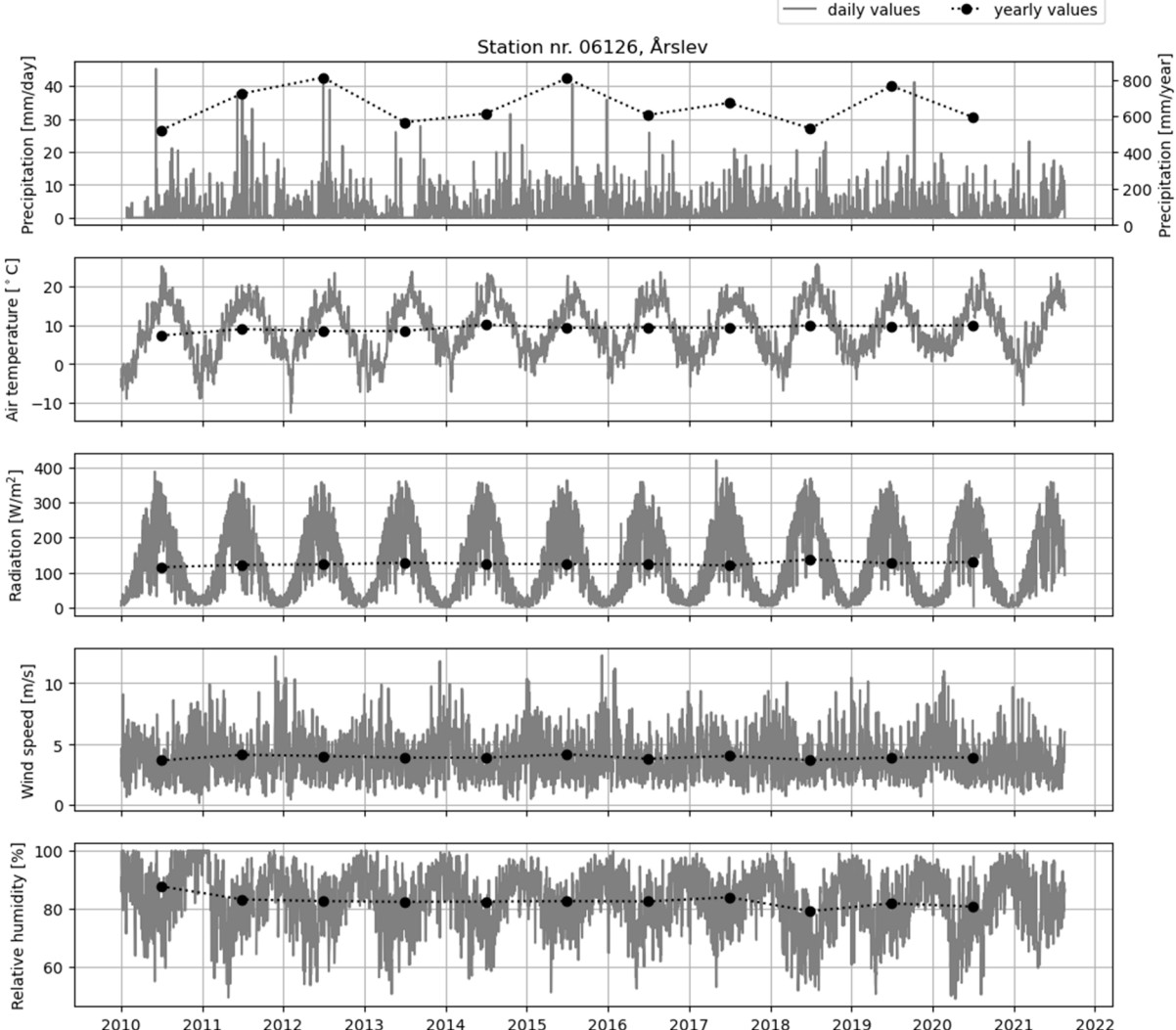

**Figure 2: Meteorological data from a weather station in Årslev app. 10 km east of the case area. Bins on the horizontal axis refer to the beginning of each year. Precipitation data is extracted in 10 min resolution and summed to daily and yearly values. Air temperature, radiation, wind speed and relative humidity are given as daily and yearly means. Based on open hydrological data from** (DMI, 2020) **where more parameters can be extracted.**

**2.3 Urban drainage system**

The urban drainage system extends from upstream Brændekilde through the town of Bellinge and further downstream meeting
the contribution from Dyrup along Odense River. The sewers were originally laid out as a combined system with domestic
wastewater and stormwater flowing in the same pipes. There are several combined sewer overflows to Odense River. Today,
most of the system is still combined, but there are also a few newer developments with separate stormwater systems with
outlets running out of the catchment area, thereby not significantly affecting the combined system. Figure 3 illustrates
conceptually how the main structures connect, and Figure 4 illustrates important hydraulic details and placement of sensors in
some of these. Further details about the structures, technical drawings along with CCTV of the interesting pipes can be found
in the data set (Pedersen et al., 2021a).

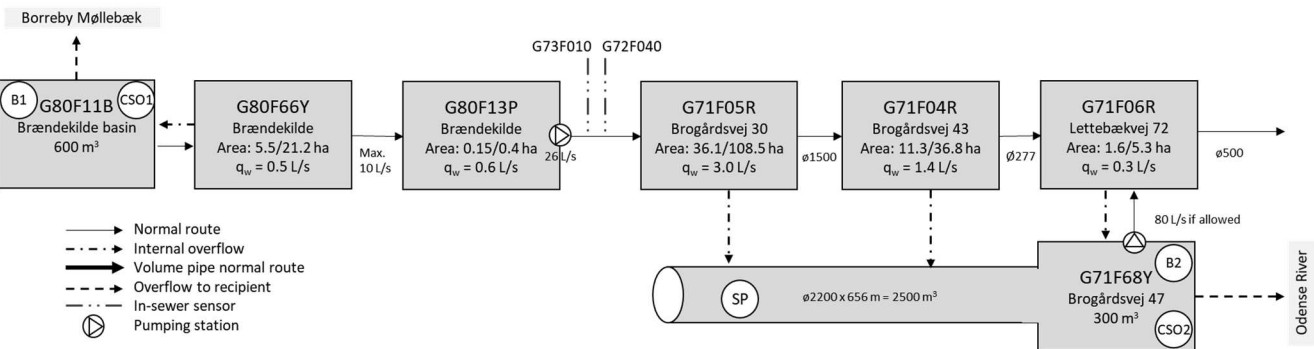

**Figure 3: Conceptual illustration of the most important hydraulic structures in the urban drainage system. "Area" indicates the impervious area/total area connected to a structure, while $q_w$ is the average daily wastewater load connected to each structure. The total average daily wastewater discharge upstream of G71F06R is 5.8 L/s (equal 501 m³/day). B (basin), CSO (combined sewer overflow), SP (storage pipe).**

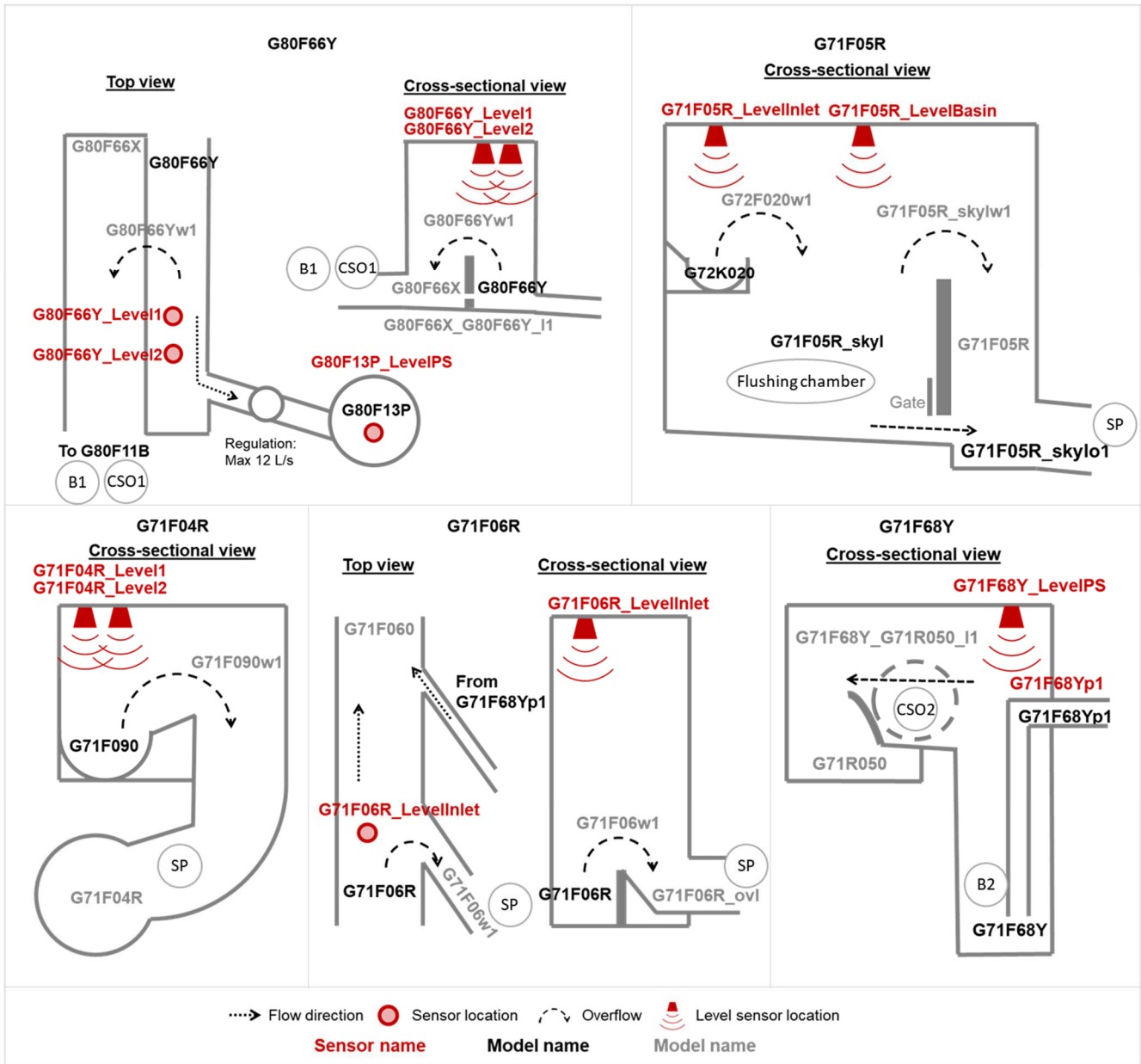

**Figure 4: Illustration of the most complex hydraulic structures with relevant names referring to nodes in the models (black text, see also Figure 1 and Figure 3) and sensors (red text, see further details in Section 3). The grey texts are names of additional model locations.**

### 2.3.1 Brændekilde

Brændekilde had its own local wastewater treatment plant until around 1990, which now serves as an open-air detention basin for combined wastewater, see Figure 5 (G80F11B, dashed white line). Combined wastewater from Brændekilde sewer branches arrives at the structure G80F66Y, from where it flows to a pumping station (G80F13P), which pumps it further

downstream to Bellinge. The pumping station also receives separate wastewater from houses in a larger surrounding rural area indicated by the red lines in Figure 1B and Figure 5 (here, the green pipe coming from north only visible). In high-flow situations water is diverted from G80F66Y to the open-air detention basin (G80F11B) through an internal overflow structure, where it is temporarily stored until it can flow back down to the pumping station. The basin has an overflow weir that discharges to a small stream called "Borreby Møllebæk", a tributary to Odense River.

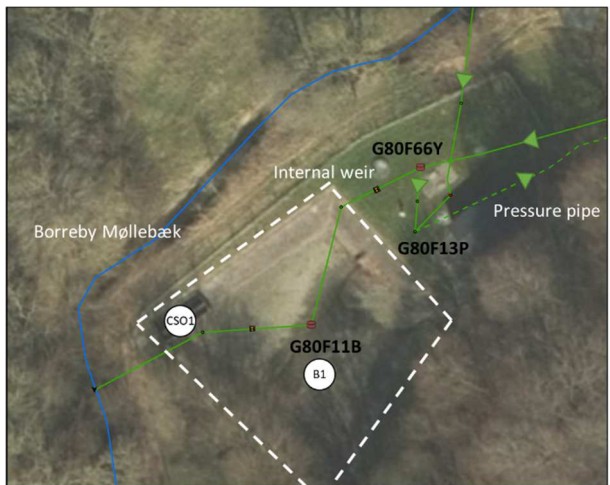

**Figure 5: Location of the hydraulic structures at Brændekilde, where observations is obtained. Basin B1 and CSO1 is located here. Background maps provided by DADSE (2020).**

### 2.3.2 Bellinge

In Bellinge, Figure 6, the system previously had problems with frequent flooding and combined sewer overflows. This led to the construction in 2010 of a large underground storage-pipe and basin, which reduced the number of overflows to Odense River to approximately five per year. Models are available for both the old system (2009, may be useful when seeking to understand the historical evolution of the system) and for the new system (from 2010, may be useful when comparing with observation data) and can be found in the data set (Pedersen et al., 2021a). The storage capacity in the storage-pipe and basin is activated during medium to large rain events via three internal overflow structures, G71F05R, G71F04R and G71F06R, located in the old combined sewer main in Bellinge (Figure 3). The storage-pipe and basin is emptied by pumping to the sewer main in G71F06R, according to a coordinating rule-based control scheme that depends on the current water level in the structures G71F05R, G71F06R and G71F68Y, but also by the current inlet flow rate at the downstream WRRF at Ejby Mølle.

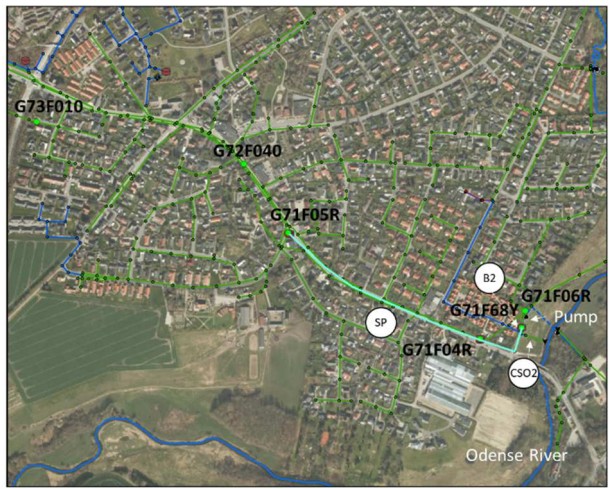

**Figure 6: Location of the sensors in Bellinge. The storage-pipe (SP) and basin B2 are highlighted (cyan) and the location of CSO 2 is indicated. Background maps provided by DADSE (2020).**

## 3 Observation data

### 3.1 In-sewer sensors

Throughout the catchment, several sensors provide data about the state of the system, see Table 1. Most sensors are level-meters, but other information is also collected. Figure 4 illustrates the location of the sensors placed in the most important or complicated structures (G80F66Y, G71F05R, G71F04R, G71F06R and G71F68Y).

The sensors in Brændekilde are in the in the basin (G80F11B), in the internal overflow structure (G80F66Y), and in the pumping station (G80F13P), see Figure 4 and Figure 5. The pumping station is fairly simple with a large manhole serving as sump and has sensors measuring the water level and power consumption.

In Bellinge, the volume pipe with a diameter of 2.2 m (ø2200 mm) received overflow from the old system from three internal overflow weirs: G71F05R, G71F04R and G71F06R. The upstream overflow structure, G71F05R, has a small flushing chamber with a storage volume and a gate (a position throttle), which closes in the beginning of a rain event in order to store water for flushing the storage-pipe in the end of an event. This is implemented in the models, however in reality the gate has an opening and closing time of 12 seconds, which the models are not able to replicate. When water overflows from the old system to this flushing storage volume, the tank quickly fills and water overflows across a weir, filling the storage-pipe and basin. When the storage-pipe has been in use and emptied again after a rain event, the subsequent flushing storage-pipe creates a small distinct peak in the water level data at the downstream basin (G71F68Y). A full chamber at G71F05R allows flushing of the storage-pipe up to three times. At G71F04R water can overflow directly from the old system into the storage-pipe. Two level sensors are located at the up- and downstream end of the weir.

The third internal weir is at G71F06R where water overflows to the basin in G71F68Y downstream of the storage-pipe. A level sensor right next to the weir allows monitoring this internal overflow. After a rain event water is pumped back to G71F06R from the G71F68Y basin. The pumping outlet is a few meters downstream of the level sensor to reduce pumping artefacts in level measurements and potential backwater effects that lead to overflow at the G71F06R weir. In G71F68Y the water from the storage-pipe is led to a deep pump sump. If the basin is full (2800 m$^3$), combined wastewater can overflow to Odense River through a rotating screen and a bendable weir. There is one level sensor in the pump sump along with a flow meter in the pressure pipe from the pumps. The pumps in G71F68Y are dry-connected pumps with a frequency converter installed controlled by the level in both the pump sump and in G71F06R. The pumps run alternating with a design capacity of max. 80 L/s.

A 1-minute temporal resolution is applied to all data from permanent sensors routinely gathered in VCS' supervisory control and data acquisition (SCADA) system, but the resolution is 2 minutes for the mobile sensors installed in G73F010 and G72F040 due to concerns about battery life.

**Table 1: Structures with sensors installed, see Figure 1 and Figure 3 for locations and Figure 4 for details about the sensor installations. Corresponding model names are given. In all structures except G73F010 and G72F040 sensors are permanently installed.**

| | Structure | Sensor name | Type | Device | Model name |
|---|---|---|---|---|---|
| **Brændekilde** | G80F11B | Level basin 1 | Level | Radar | G80F11B |
| | | Level basin 2 | Level | Radar | G80F11B |
| | G80F66Y | Level inlet 1 | Level | Radar | G80F66Y |
| | | Level inlet 2 | Level | Radar | G80F66Y |
| | G80F13P | Level pump sump | Level | Transducer | G80F13P |
| | | Power 1 | Power | | G80F13Pp1 |
| | | Power 2 | Power | | G80F13Pp1 |
| **Bellinge** | G71F05R | Level inlet | Level | Radar | G72K020 |
| | | Level basin | Level | Radar | G71F05R_skyl |
| | | Position throttle | Position | | G71F05R_skylo1 |
| | G71F04R | Level inlet 1 | Level | Radar | G71F090 |
| | | Level inlet 2 | Level | Radar | G71F090 |
| | G71F06R | Level inlet | Level | Radar | G71F06R |
| | G71F68Y | Level PS | Level | Radar | G71F68Y |
| | | Flow pump | Flow | | G71F68Yp1 |
| | | Power 1 | Power | | G71F68Yp1 |
| | | Power 2 | Power | | G71F68Yp1 |
| | G73F010 | | Level | Ultrasonic | G73F010 |
| | G72F040 | | Level | Ultrasonic | G72F040 |

### 3.2 Rainfall data

Two rain gauges with more than 10 years of observations available are located in and just outside the catchment ("5425 Brændekilde" and "5427 Dalum") see Figure 1A. These gauges are a part of a national network of utility-owned rain gauges, which undergo quality control by the Danish Metrological Institute (DMI) (Jørgensen et al., 1998). They are tipping bucket
gauges and measure the number of tips every minute, which is then converted to rainfall intensity. VCS also temporarily installed a tipping bucket rain gauge at the center of the catchment for a one-year period, see "Aabakken" on Figure 1A. VCS has also operated a local X-band weather radar since 2012, and time series from this radar (1 min resolution) are provided for the area of interest in cell sizes of app. 925x925 m, see Figure 1A. The radar is dynamically adjusted using app. 10 rain gauges located in the area covered by the radar according to the method described in (Borup et al., 2016). As a supplement VCS has
also since 2017 adjusted the signal from DMI's C-band radar (5 min resolution) located in Virring app. 80 km away with the same methods, in order to cover parts of VCS's service area which are not entirely covered by the X-band radar. However, we note that the distance of 80 km to the C-band radar is close to the limit for which a radar of this kind can deliver reasonably accurate quantitative rainfall estimates (Thorndahl et al., 2014).

### 3.3 Availability of rainfall and in-sewer observation data

Figure 7 illustrates the temporal availability of the data provided, including both the in-sewer sensor data (cf. Table 1) and the rainfall data products. The downtimes of each sensor are not long, with the exception of one of the level sensors at G71F04R. During 2019, VCS installed more permanent sensors throughout the urban drainage system, which is the case for Brændekilde (G80F11B and G80F66Y).

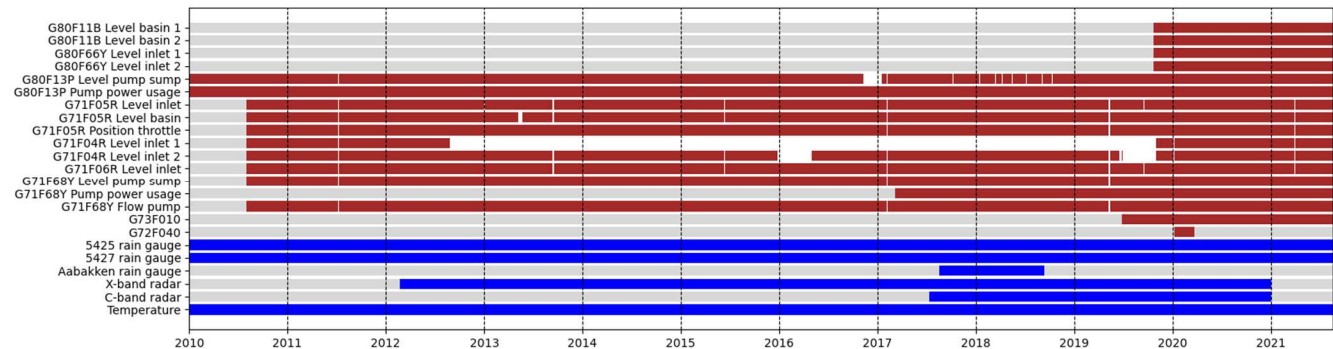

**Figure 7: Time periods with in-sewer sensor data (red bars) and rainfall observation data (blue bars) provided. White bars indicate periods with erroneous sensor data (downtimes), and grey bars indicates periods without data. Power for the pumping stations and the weather radars has not been checked for erroneous data. G73F010 and G72F040 are level sensors.**

All observation data is provided in the data set (Pedersen et al., 2021a) and in most cases available as uninterrupted data series since 2010 (updated in August 2021, i.e. more than 10 years). VCS mainly uses sensor data for day-to-day operation and
planning of maintenance work, but also for developing a digital twin strategy (Pedersen et al., 2021b). In cases where a sensor

has limited effect on the system control, longer periods with missing data may occur as those data are not a high priority (e.g., one of the duplicate sensors at G71F04R). Exact documentation of sensor maintenance has not been a high priority over all the years, and it is therefore presently not possible to give an overview of when and where sensors have been repaired, replaced or received some sort of maintenance. Data is generally good, but as for all large real-world data sets, there will be errors and

anomalies. One example is a gradual increase in daily minimum water level at G71F06R from 2010 to 2015, followed by a sudden drop from in May 2015 and a sudden increase again in December, see Figure 8. Comparing photos taken in 2019 and CCTV recordings from 2013 revealed that the banquettes of the structure had been retrofitted during this period. Further investigating this by interviewing operational staff at VCS revealed that indeed, one remembered that approximately in 2015 the banquettes were retrofitted with new material. This illustrates that data can be hard to understand, but that there might be

a logical explanation behind large anomalies; in this case potentially an effect of gradual deterioration until May 2015 followed by more stationary conditions at lower water levels. Direct use of the data during low flow situation therefore has to be done with care up to this date. The behaviour in December 2015 until March 2016 could be a consequence of a very rainy season in the general Odense area during these months, which caused a lot of infiltration.

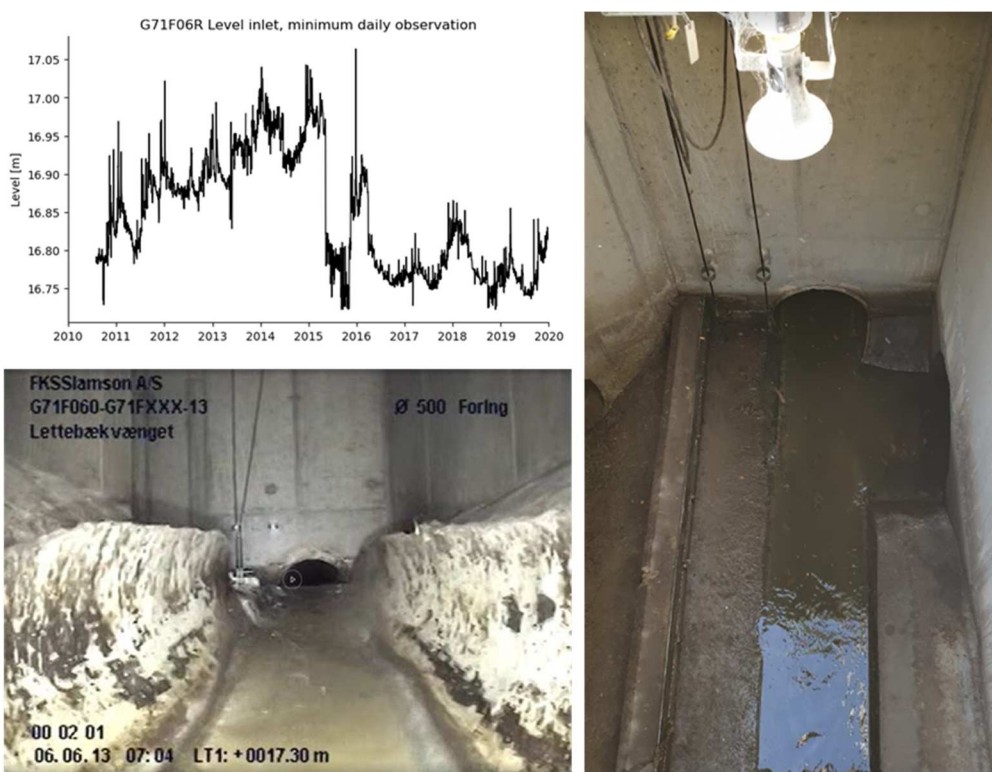

**Figure 8: Anomaly in level data and possible causes. The minimum daily water level in G71F06R (upper left corner) along with a CCTV image (lower left corner) of G71F06R from June 2013 and a photo of the same location from 2019 (right) showing also the location of the level sensor (radar). Videos and additional photos can be found in the data set (Pedersen et al., 2021a). A large anomaly in data occurs May 11th, 2015, which is possibly due to retrofitting of the banquettes in the structure at this date. Pictures by (VCS Denmark, 2020)**

The sudden drop in daily minimum water levels in 2015 is a large anomaly that is easily spotted in the data. Although the data has been analysed by the authors of this study, there might still be small artefacts in the data due to minor undocumented physical changes to the system. Small changes to the control settings of e.g., pumps have also not been thoroughly documented and can thus also appear. We acknowledge that this is not optimal. The utility company is in a transition process of changing the way meta data is logged. By exploiting the procedures in a typical Danish utility company we can hopefully start a
discussion of how to make best practices.

## 3.4 Data-cleaning

The observations from the in-sewer sensors are provided both as raw data and as a cleaned version where erroneous data points have been removed. The cleaned data were converted in time to UTC+0, see python scripts in the data set (Pedersen et al., 2021a) for details. The depth recorded by the level sensors needs to be comparable with the model results and were therefore
converted to level by adding the invert level or the 0-point for the sensor to the measured depth, see Figure 9.

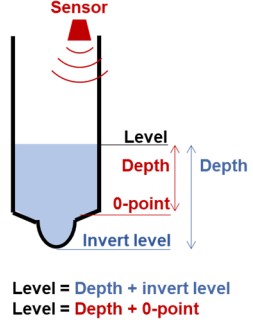

**Figure 9: The link between an observed depth, converted to level by adding a 0-point (red text), and the modelled depth and level from a model (blue text).**

VCS changed supplier of SCADA system during 2020, going from System2000 (Frontmatec) to iFix (GE Digital), which will
give a different output format. A third format will be given from the interim sensors, which is supplied from a company called Danova (Danova). For this release, it was for practical reasons, decided to use an initial set of common, simple data cleaning techniques and leave a more comprehensive data validation as a research opportunity for ourselves or others in the future e.g., (Leigh et al., 2019). The cleaning techniques included five techniques for replacing clearly erroneous observation data with Not-a-Number (NaN) values:

- *Manufacturer quality stamp*: Data stamped with "low quality" in the iFix SCADA system.
- *Manual remove:* Data that for some reason was deemed untrustworthy, for instance observation values during maintenance or start-up periods.
- *Out of bounds*: Data outside a defined physically meaningful range of possible values (e.g., bottom and top levels of a pipe/basin).
- *Frozen sensor*: Data not changing during a time period of e.g., 20 minutes.

- *Outlier:* Data with spikes with a manually chosen height and duration; in our case only applicable for interim Danova sensor data, which occasionally showed spike patterns which are probably not correct.

Data from iFix is set to reduce data storage requirements at the sensor by leaving out observations that changes in the coming timestep below a given threshold (for water level sensors most often set to 1 cm). Therefore, this should not be seen as a period of failure, or signal loss, and for this script forward-filling is applied to the values which have these properties.

Table 2 shows the number of data points flagged by each error function, which is also illustrated in Figure 7. Simple gap-filling based on linear interpolation was done for gaps shorter than 5 minutes, as increased gap-filling period would increase the risk of interpolate a potential peak. Python scripts for the data cleaning and gap filling are given in the data set (Pedersen et al., 2021a).

**Table 2: Overview of the five error types in data from the different sensors. Values are one-minute values.**

| Structure | Sensor name | Time steps | Stamped error | Manual remove | Out of bound | Frozen normal | Outlier |
|-----------|-------------|-----------|---------------|---------------|--------------|---------------|---------|
| G80F11B | Level basin 1 | 956.975 | 46 | 0 | 0 | 0 | - |
|  | Level basin 2 | 956.942 | 46 | 0 | 0 | 0 | - |
| G80F66Y | Level inlet 1 | 958.160 | 36 | 0 | 38 | 0 | - |
|  | Level inlet 2 | 958.160 | 36 | 0 | 39 | 0 | - |
| G80F13P | Level pump sump | 6.115.561 | 57 | 0 | 353 | 1.224 | - |
| G71F05R | Level inlet | 5.810.282 | 53 | 0 | 4.533 | 28.531 | - |
|  | Level basin | 5.810.282 | 53 | 0 | 21.651 | 38.203 | - |
|  | Position throttle | 5.810.282 | 53 | 0 | 1 | 0 | - |
| G71F04R | Level inlet 1 | 5.810.603 | 503 | 3.222.281 | 3.932.754 | 3.487 | - |
|  | Level inlet 2 | 5.810.603 | 503 | 69 | 231.610 | 204.539 | - |
| G71F06R | Level inlet | 5.810.282 | 54 | 6.120 | 4.428 | 23.755 | - |
| G71F68Y | Level pump sump | 5.809.728 | 12 | 6.120 | 7.470 | 0 | - |
|  | Flow pump | 5.809.298 | 20 | 6.120 | 0 | 15.214 | - |
| G73F010 |  | 1.128.961 | 0 | 0 | 0 | 0 | 4 |
| G72F040 |  | 107.685 | 0 | 0 | 103 | 0 | 0 |

## 4 Simulation models

Physically based, distributed urban drainage models, constructed in software packages such as MU (DHI, 2020) and SWMM (EPA, 2020) that are used here, are the most detailed type of urban drainage system models available and contain two main components: a surface module that calculates rainfall-runoff from each sub-catchment to the pipe system and a hydrodynamic model that calculates the flow in the pipe system. The hydrodynamic model solves the St. Venant equations across the pipe network and represents head loss in manholes and flow in overflow weirs and other hydraulic structures using standard hydraulic equations. The surface modules are in principle lumped-conceptual, but the sub-catchments are distributed in space

according to the overall layout of the pipe network. The detailed mathematical formulations and numerical schemes used are different in MU and SWMM, and model users may choose between several options for describing especially the lumped-conceptual surface runoff model components. Rainfall data and wastewater loads are the main model forcings, but e.g., infiltration-inflow and pumped flows can be used as additional forcings. The main model attributes are surface areas, imperviousness and hydrological response time of sub-catchments, and the main asset data are features of the pipe network (diameter, length, roughness) and hydraulic structures (basin volumes, weir data, etc.). Several model forcings and attributes may be determined from independent external data sources or be considered parameters that can be calibrated based on observation data from the system. VCS uses MU in the daily modelling and model updating work, which is however not easily accessed by potential users of the data set because of its proprietary nature. We therefore also provide a SWMM model (created to mimic the behaviour of the MU model), which is open source and thus readily available for use by the international research community.

## 4.1 MU-models: building parts

The distributed urban drainage model is made by VCS in the MU software system (DHI, 2020) and is part of an operation model that is run and compared with obervations on a routine basis as part of a digital twin environment currently under development (Pedersen et al., 2021b). The hydrodynamic model consists of around 1000 nodes and 51 km of pipes (40 km combined sewer pipes, 7 km separate stormwater pipes, and 4 km wastewater pipes, see Figure 1B), and the surface module consists of 713 individual sub-catchments with sizes up to 10 ha with median size on 0.3 ha. The downstream model boundary (model outlet) is chosen so that there are no backwater effects at any of the sensor locations in Bellinge from the downstream parts of Odense city. Dyrup is only in the model to ensure that the effect from this part is considered in the main pipe downstream of G71F06R. The emptying of the basin in Bellinge, G71F68Y, is in reality controlled by local regulatory and coordinating rules as well as global system-wide rules according to all the basins in the entire network of Odense and Ejby Mølle WRRF. The local rules are specified in the models, but overriding signals from the global system control is not considered here. The model can therefore not empty the basin in Bellinge realistically for some periods depending on the filling degree of other basins close to the WRRF. As the MU model software cannot handle frequency converting pumps, the exact modelling of the pumping curves can furthermore be challenging. The screen and bendable weir in G71F68Y are described in the model as a regulated pipe with a specific QH curve corresponding to the detailed characteristics of these elements.

Calibration of urban drainage models has been subject to a lot of research internationally for more than a decade (e.g. Bach et al. (2014), Broekhuizen et al. (2020), Nagel et al. (2020), Tscheikner-Gratl et al. (2016) and Vezzaro et al. (2013)). However, VCS has a philosophy of transparency in models, where understanding the system behaviour is considered more important than ensuring a perfect calibration with non-transparent parameter sets, meaning that VCS prefers not to tune conceptual parameters to unrealistic values in order to fit models to observations. Therefore, the various parameters of the model were not calibrated, and standardized parameter set were used when possible. The model has however frequently in the past been

validated against observations from the system, and the causes of a poor fit have been investigated and corrected. This could for instance be an error in the registration of the level of an overflow weir, or an impervious area connected differently to the system than anticipated. When discovering such errors, the system data was corrected in the asset database and the model was updated according to the asset database.

The forcings and physical attributes of the system implemented in the provided MU model are outlined in Table 3. VCS has experienced that the current model for imperviousness overestimates the rainfall-runoff from some of the impervious areas in the outskirt communities. These are probably not often connected to the urban drainage network and instead stormwater is infiltrationed in trenches. The imperviousness of these areas are, however, not changed until field work has shown which areas should not contribute. An internal report from VCS assessed, based on analysing data from pumping stations, that approximately 30% of the hydraulic load to Ejby Mølle WRRF is infiltration-inflow. Half of this is expected to be caused by infiltration due to cracks in pipes and manholes and the other half due to agricultural drainage pipes, which historically have been connected to the urban drainage system. Several attempts have been made in VCS to model the infiltration-inflow, for example machine-learning techniques applied to observations near the treatment plant. These can be used for estimation of the inflow to the treatment plant but seldom matches reality when scaled to upstream catchments. Therefore, infiltration-inflow was not included in the MU model provided here but we encourage potential users of the data set to investigate this further.

**Table 3. Determination of model forcings and attributes in the implemented MU model from independent data soruces.**

| Forcing/attribute | Description |
|---|---|
| Wastewater loads | Calculated based on the yearly metered water consumption for each household in a given year, converted to an average daily wastewater load and aggregated for each sub-cathment (to respect data privacy requirements). Daily demand patterns (identical for all days of the week) were based on detailed metering from district metering area (DMA) smart meter of water consumption. |
| Imperviousness | Based on satellite data from which different types of land use were identified. An assumed percentage of imperviousness for each land use type was applied to calculate an average imperviousness for each sub-catchment, as explained in the data set (Pedersen et al., 2021a). |
| Time of concentration | Assessed according to a simple classification of the size of the area, referring to the documentation in the data set (Pedersen et al., 2021a). |
| Network data | Extracted from a continuously updated asset database of pipes and manholes (Pedersen et al., 2021a). Pipe and manhole characteristics can be trusted to a high degree |
| Pipe roughness and manhole head loss | Manning numbers and parameters governing headlosses in manholes were assessed with simple rules without using exact knowledge of the system. Manning numbers were set according to the |

pipe material used (concrete, plastic and iron) and not the condition of the pipes, and every manhole was assigned the same headloss parameters.

## 4.2 SWMM: Conversion from MU

A SWMM model (EPA, 2020) was constructed based on the utility company's MU model, and validated against the MU model results. Minor modifications were applied to the SWMM model to produce results that are as similar to MU as possible. Model structure differences between MU and SWMM are highlighted in the following.

**Surface runoff.** The MU model uses a time-area runoff model (DHI, 2020), called "runoff model A", which calculates the rainfall-runoff from each subcatchment based on its area, imperviousness and time of concentration (Table 3) and initial loss (set to a default value). SWMM has another way of estimating surface runoff with some similar attributes (area, imperviousness, initial loss) but with other attributes than the time of concentration describing the runoff routing (width, slope and Manning coefficients) for both impervious and pervious areas (Rossman and Huber, 2015). In Denmark, there has not been a tradition for including runoff from pervious surfaces in urban drainage models, except very recently as the occurrence of large cloudburst events has increased. In order to make the two models as similar as possible, the parameters for pervious surfaces in SWMM's infiltration module were thus set to unrealistically high values, so that rainfall on such surfaces readily infiltrates into the ground instead of producing runoff to the urban drainage system. On impervious surfaces, the parameters were set to produce run-off similar to the runoff simutated with the MU model.

**Network.** The hydraulic calculations of the pipe network are quite similar in both models, solving the full Saint-Venant equations. However, the calculation of headloss in manholes is different, as MU takes their volume into account whereas SWMM neglects it. In order to obtain more similar models, the largest manholes were in SWMM represented as "storage units" with designated volumes. In MU there is a default setting where the length of pipes shorter than 10 meter is adjusted to 10 meter for computational stability. This is not the case in SWMM. MU also generates tiny amounts of water in empty pipes to avoid zero values (again for numerical stability). This is an issue in the large storage-pipe, which is empty most of the time, and it was circumvented for this study by inserting a fictitious outlet in the MU model in the normally dry storage-pipe, allowing water to disappear when the water level in the storage basin is low. SWMM does not generate water in dry pipes and therefore a fictitious outlet was not inserted. In Denmark, a special non-circular shape of pipe has historically been used; MU includes this as a standard cross section type while SWMM does not, and a custom cross section was therefore defined for the relevant pipes in the SWMM model, see the data set (Pedersen et al., 2021a) for further information. A more thorough comparison of the two simulation softwares can be found in (Borah, 2011) where different model formulations used in typical software products, including both SWMM and MU, are described.

# 5 Model and data comparisons

In the following section we illustrate the nature of some of the data, the behaviour of the system as well as the performance of the two models. The models' response and the observations can be compared at the sensor locations (see Table 1) and as stated in the data set (Pedersen et al., 2021a). The analyses in this section will focus on two types of comparisons for three selected rain events: a comparison of the two simulation models discussing structural model uncertainty potentially leading to significant differences, and a comparisons of selected model results with in-sewer observation data discussing how well the models can represent the dynamics in the urban drainage system given the uncertainty of the inputs.

## 5.1 Selection of events

Data for three rain events of short duration and high intensity were selected from the 10-years of observations to illustrate the dynamics of the system. These three events (from June 2012, August 2015 and August 2018) were selected due to similar large rain depths and dynamics in both of the two permanent rain gauges (5417 and 5425) located on opposite sides of the catchment; and X-band radar rainfall data was furthermore available for the third event (Table 4). Similar measurements in the two gauges indicates that there is little spatial variation in rainfall rates, which reduces the risk of poor spatial sampling of the rainfall and makes the comparison of MU and SWMM responses with in-sewer sensor data simpler. Intensity-duration-frequency (IFD) characteristics of the three events are illustrated in Figure 10, with the national regional IDF curves from Odense as background for comparison (Gregersen et al., 2013, 2014, 2016; Madsen et al., 2017).

**Table 4: Chosen events and their characteristics.**

| Event no. | Event date | Volume [mm] 5425 | 5427 | Duration [min] 5425 | 5427 |
|---|---|---|---|---|---|
| 1 | 29-06-2012 | 19.4 | 23.6 | 259 | 257 |
| 2 | 31-08-2015 | 22 | 25.6 | 84 | 103 |
| 3 | 11-08-2018 | 6.6 | 9.6 | 92 | 144 |
| | 11-08-2018 (x-band radar) | 0-9 mm | | | |

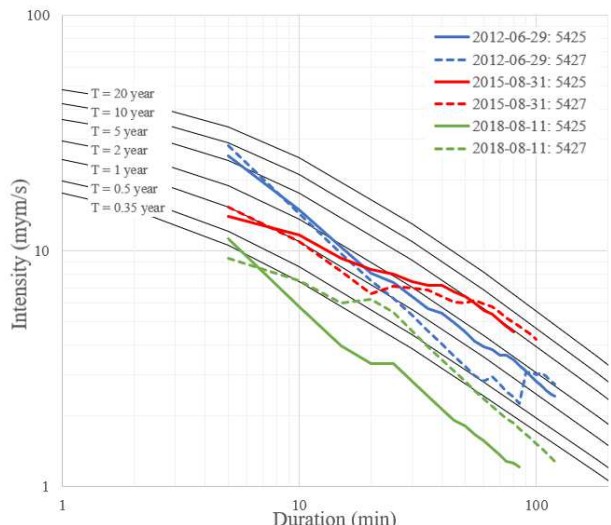

**Figure 10: Intensity-duration-frequency (IDF) characteristics for the three selected rain events (2012, 2015 and 2018) measured at two different rain gauges (gauge 5425 and 5427), plotted on top of national regional IDF curves for Odense (Gregersen et al. (2014)). T indicates return periods, corresponding to the reciprocal of the frequency.**

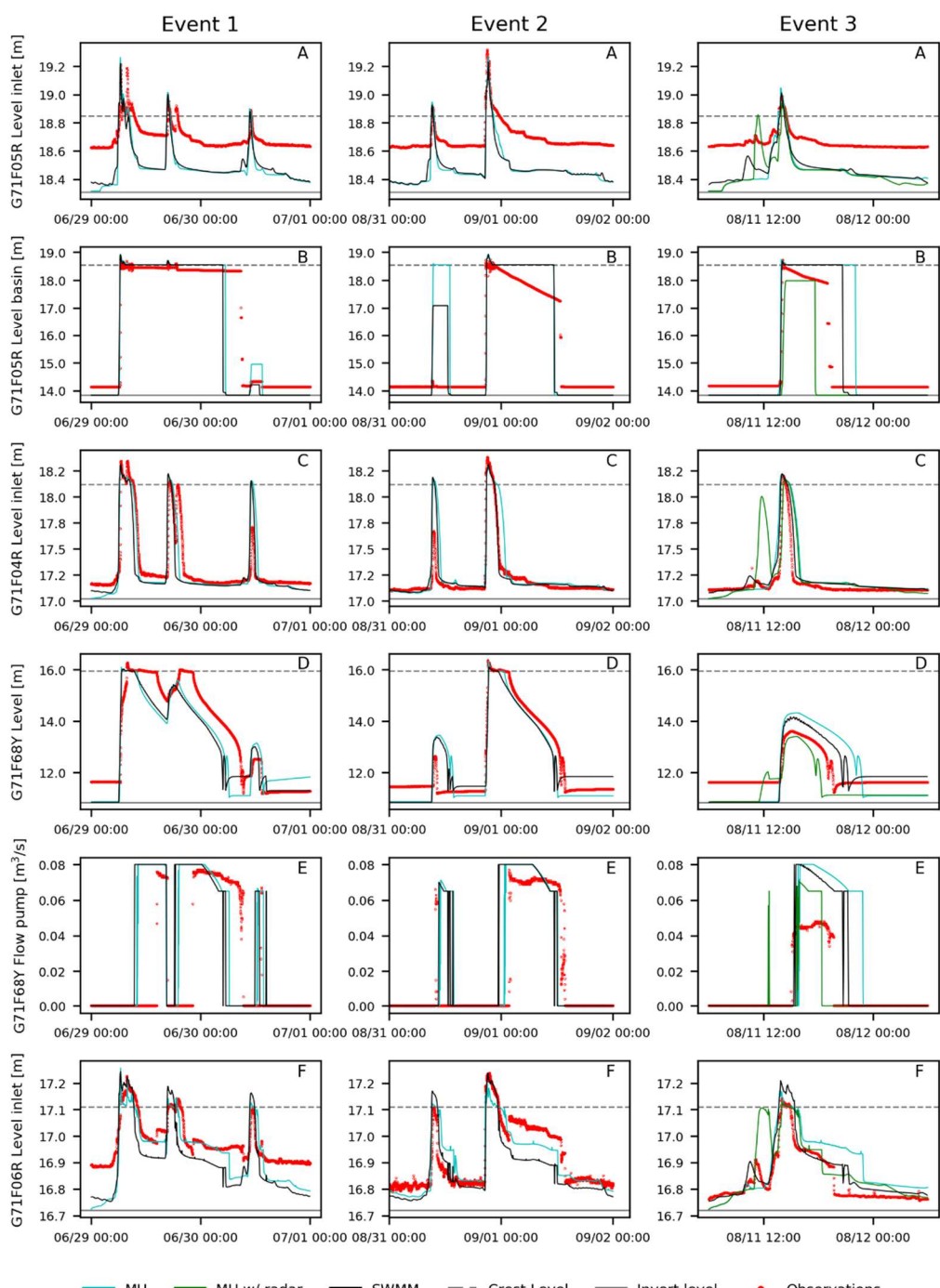

**Figure 11: Computed time series of level and flow using MU (black) and SWMM (green) compared with observations (red dotted) for the sensors in and around the storage-pipe in Bellinge. Grey dotted horizontal lines indicate weir crest levels. The green time series for event 3 is MU with radar input. Note that the vertical axes are not the same between for all the rows of the figure. Water level units are meters above sea level, while flow units are m³/s.**

**5.2 MU vs SWMM**

The model results for the three events are shown in Figure 11 together with water level observations for six of the water level sensors located in structures in Bellinge that are important to the dynamics of the storage-pipe and basin (G71F05R, G71F04R, G71F06R and G71F68Y). Generally, the two models show the same tendencies in runoff response to rainfall, but there are small differences in the timing of flow in the most upstream nodes of the system (not shown) due to the different surface runoff models implemented in MU and SWMM. The integrating effect of the overall sewer catchment, however, means that the model

results are very similar in the measurement locations that are located further downstream. An exception is in G71F06R (Figure 11F) where the peaks stay higher for longer in the SWMM model for all of three events. A throttle pipe is located immediately upstream from this sensor (ø277, Figure 3), and through inspections of the model it was found that SWMM estimates up to 50% more flow through the throttle than MU in the peak situations, thereby allowing the peak level in G71F06R to be higher for longer time. This is due to the conceptual difference in manhole representation between the models, where MU has a

volume and a head loss in the manhole while SWMM does not. It is not known which of the solutions that is most accurate.

**5.3 Model results vs observations**

The observations in G71F05R Level inlet (Figure 11A) have a higher base level than the model results. Water depths lower than 25 cm above the invert level were not recorded here during the 10 years measurement period. The pipe has a very low gradient and therefore the elevated water level may be due to some downstream partial blockage or dislocation of a pipe. The

level observations in the flushing chamber (G71F05R level basin, Figure 11B) show a slow decrease after the chamber has been filled up, especially for event 2 and 3. None of the models show this, which seems to indicate a small leak during some events from the gate that holds water back in the chamber. The model results in G71F04R Level inlet (Figure 11C) are mostly similar, except that MU maintains a higher water level slightly longer than SWMM does. In the G71F68Y basin the filling and emptying dynamics are generally very good (Figure 11D). When the basin has been emptied after a rain event, a small peak

occurs as the gate in G71F05R is opened and the volume pipe is flushed. The water level for event 1 is slightly underestimated by both models, while it is overestimated for event 3 (Figure 11D). The fact that both models are biased for these events suggests that the rainfall input might also be biased. Radar observations from the X-band radar were therefore used as input to the MU model for event 3. This led to a slight underestimation of the modelled peak water levels (radar) in most locations (Figure 11A, B, C, D, event 3). Despite the availability of two rain gauges on opposite sides of the catchment and a nearby

radar, there is still a considerable uncertainty in the rainfall input. The pump rate G71F68Yp1 (Figure 11E) is well simulated for event 2 (Figure 11E), but it is overestimated for event 3 (Figure 11E, event 3). Global control settings might also be responsible for the fact that the pumping starts earlier in both of the models than in the observations (Figure 11E).

Both the model results and the observations show that the emptying of the G71F68Y basin downstream of G71F06R starts

after the overflow in G71F06R has ended (Figure 11F). Event 2 contain larger increases in water level at G71F06R than event

1 (Figure 11F). The pumping effect on the downstream water levels are thus not always similar despite nearly identical pumping flows. MU and SWMM also disagree on the size of this effect despite identical pipe geometries and pumping characteristics. The sudden 2015-drop in the dry weather water level at G71F06R in event 2 that was presented in Figure 5 is also seen between the first and the two other events (Figure 11F).


Several of the sensors are not placed directly above the point in a manhole/pipe/storage tank with the lowest invert elevation (Figure 9). In these locations, it is not possible to measure water levels below 0-level of the sensor, leading to an offset in the measured values that has to be accounted for. For most of the sensor locations, the offset is very small in the order of a few centimetres. It is, however, visible for both G71F05R Level Inlet (25 cm, Figure 11A) and G71F68Y Level (23 cm, Figure

11D), where the sensor data cannot reach the lowest values of the simulations. As also shown in Figure 11F there is a gap in base level in G71F06R for the first event, which is due to the gradual increase in minimum daily water level as shown in Figure 8.

An acceptable agreement between measured and simulated values is generally shown for the sensor locations and the three

selected events. More sensors were however installed at two locations in Brændekilde in fall 2019, see Figure 7. Observations and MU simulation results for a single event are shown for G71F66Y in Figure 12. This simulated level is lower than the observations, which suggests that the models overestimate surface runoff in the outskirts areas and highlights that the models, although mostly physically based, have not been calibrated against observations. This illustrates some of the uncertainty in model attributes, which may be addressed in further research using the provided data and models.

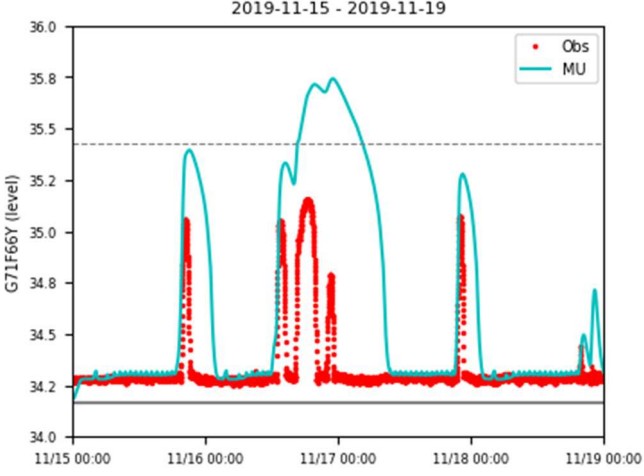


**Figure 12: MU model results (blue) and observations (red dotted) at the internal overflow structure G71F66Y in Brændekilde. The grey dashed horizontal line indicates weir crest level. The rain event 16-11-2019 15:39 (second peak) consisted of 6.2 mm rain in 426 min (second peak), while the other peak periods had lower volume of rain.**

**6 Potential use of the data set**

We envision that the data set presented in this paper can be used for a large span of research areas and problems such as: (i) automated error detection and gap-filling in data series, (ii) multi-site comprehensive data validation using e.g., machine learning and artificial intelligence, (iii) development of surrogate models of hydrodynamic models as well as entirely data driven models, (iv) development of better models components for physically-based hydrodynamic models, (v) developing better lumped-conceptual model parts describing rainfall-runoff from pervious areas and infiltration-inflow in pipe systems,

(vi) use of satellite data to improve the surface catchment characterisation and potentially discover flooded areas and nodes, and (vii) data assimilation for real-time modelling, forecasting for warning and control, etc. Furthermore, supplementary data such as CCTV not discussed in detail here may be used to check assumptions about pipes, manholes and hydraulic structures in future work with the provided data and models.

The data set is currently used by the authors to identify sources of model uncertainty (Pedersen et al., submitted), anomaly detection of observations using machine learning techniques (Palmitessa et al., in preparation), and for teaching activities in urban drainage at DTU Environment. With increased focus on digitalisation, the data set can also be used to initiate discussions on data acquisition and transfer needs (Eggimann et al., 2017) in order to gain insight into urban drainage systems that are gradually becoming more complex (Blumensaat et al., 2019), and to initiate discussion about what metadata and logs should

be stored to ensure available information for future use. The water sector is furthermore known for inadvertently "hiding" data in silos hosted both within utilities (e.g. in different departmental systems) and by different external contractors, which makes integrated analysis tedious and resource demanding (e.g. Lund et al. (2021)). We thus also encourage discussion on how the various information sources provided here may work together as required in future digital twin environments (Pedersen et al., 2021b).


We hope that the data set can also be an inspiration for how to manage data and models for utility companies across the globe. The utility company, VCS, in particular hopes to benefit from future research exploring the here presented data set, motivated by the availability of state-of-the-art models and the uniqueness of the long observation period, and to inspire discussions with other utilities that share common goals of improving performance through high-quality data acquisition and modelling.

**7 Data availability**


The data are available from the DTU Data repository https://doi.org/10.11583/DTU.c.5029124 (Pedersen et al., 2021a) and consist of the following items (cf. Figure 13).

**Asset database – urban drainage system**: The asset database of the system of Bellinge with information of how it is registered by manholes and links. The links from a database extraction from 2007 is also provided to illustrate the system before the

storage pipe and basin was built.

**Sensor data – urban drainage system**: Sensor data measured in the system are located here both as the raw data extracted from the SCADA system and as cleaned data, where data is checked for simple errors.

**Rain gauge data**: Data from the permanently installed rain gauges 5425 and 5427 along with the temporally rain gauge Aabakken are in this folder.

**Radar data etc.**: This item contain the radar data from both X-band and C-band radar. Besides that, meteorological data from the weather station in Årslev is located here.

**Drawings**: As-built drawings of the structure is located here together with photos taken in 2019.

**CCTV – urban drainage system**: CCTV videos are saved in this item of selected stretches to indicate how the system looks within the pipes.

**Orthophoto, Digital Terrain model etc.**: Data from DADSE are in this folder, where digital terrain models, orthophotos and maps of the areas can be found.

**Models**: MU and SWMM models of the current system are in this item. Besides that, a MU model of Bellinge in 2009 has been made, in case some readers are seek information about the system prior to 2009 when the basin and the volume pipe was built.

**Catchment description**: This item gives the catchment description, indicating the different classes that the imperviousness percentage is based upon.

**Scripts**: To ensure a certain data quality data can be prepared with the scripts from this folder

The data set is split into 9 items as there is no need to download all for a very specific use. Figure 13 gives an overview of the
data repository with clear identification of both the ownership of the data and how the data would normally be accessed, prior to publishing this data article and repository. The provided data comes with a Creative Commons license CC BY 4.0, except from some of the rain data from DMI which comes with a CC BY NC 4.0 license (commercial use not permitted).

| Folder | Subfolder | Data | |
|---|---|---|---|
| #1 Asset data | | Links.shp | 🔒 |
| | | Manholes.shp | 🔒 |
| | | Links_2007.shp | 🔒 |
| #2 Sensor data | 2 Cleaned data | In-sewer sensors | 🔒 |
| | 2 Raw data | In-sewer sensors | 🔒 |
| | | Temperature at WRRF | 🔒 |
| #3a Rain gauge data | | 5425 - files    CC BY-NC 4.0 | $ |
| | | 5427 - files    CC BY-NC 4.0 | $ |
| | | Aabakken - files | 🔒 |
| #3b Radar data etc. | 3b DMI Meteorological station | DMI Meteorological station | 🔓 |
| | 3b DMI C-band | DMI C-band | $ |
| | 3b Local X-band | Local X-band | 🔒 |
| #4 Drawings | 4 Drawings | Drawings | 🔒 |
| | 4 Photos | Photos | 🔒 |
| #5 CCTV | | CCTV | 🔒 |
| #6 Orthofoto, DTM etc. | 6 Digital Height model | DHM Height Curves | 🔓 |
| | 6 Digital Height model (rain) | DHM Rain | 🔓 |
| | 6 Background map 25 | DTKmap25 | 🔓 |
| | 6 Background map 100 | DTKmap100 | 🔓 |
| | 6 Geographical information | GeoDanmark | 🔓 |
| | 6 Orthophoto | GeoDanmark orthophoto | 🔓 |
| #7 Models | 7 SWMM | SWMM | |
| | 7 Mike Urban | Mike Urban - files | 🔒 |
| | | Mike Urban+ - files | 🔒 |
| | 7 Mike Urban anno 2009 | Mike Urban anno 2009 | 🔒 |
| | | Mike Urban+ - files | 🔒 |
| #8 Catchment description | 8 Land usage | OV-2016_NF-2013_v8.tif | 🔒 |
| | 8 Catchment areas + imp. | Samlet_feb2019a_Bellinge.shp | 🔒 |
| #9 Scripts | | Scripts | |

Access:
- 🔒 Normally private data
- 🔓 Third-party open access data
- $ Normally third-party subscription data

CC BY-NC 4.0
License type. Data must not be used for commercial purposes.

Other data has no additional restriction other than appropriate credit (CC BY 4.0).

Ownership:
- SDFE
- VCS Denmark
- DMI
- Developed for this study

**Figure 13: Overview of the items in the data repository folders and their subfolders, with an indication of the ownership of the specific data and how these data sources are normally accessed. The sensor and rain data in Figure 7 can be found in the folders "#2 Sensor data", "#3a Rain gauge data" and "#3b Radar data".**

## 8 Conclusion

Open-access data and models are currently non-existent within the urban hydrology and urban drainage research community. This comprehensive release of data from a real urban drainage system serving a 1.72 km$^2$ area in the town Bellinge near Odense, Denmark, includes more than 10 years (2010–2020) of rainfall data from rain gauges, meteorological data from a nearby weather station, and level and flow data from in-sewer sensors. In addition, 8 and 3 years of data from X-band and C-band weather radars are provided, as well as two near-identical hydrodynamic distributed urban drainage models constructed in the software tools Mike Urban and EPA SWMM. This case is well-suited for research within a broad range of topics such as data quality control and optimal sensor maintenance, automatic error and anomaly detection, model calibration, uncertainty analysis, development of surrogate models, data-driven modelling, forecasting, real-time control, digital twins, etc. We hope the community will adopt the Bellinge case as a benchmark that enables independent testing and replication of results from future scientific developments. The case should also be highly relevant for teaching purposes.

This comprehensive data set provides a unique opportunity to explore several aspects of urban drainage systems and to publish research that can be replicated by others. The two urban drainage models respond almost identically to rainfall forcing; however, they are not calibrated to the observations, which is most evident for sensors located upstream. The models have the local regulatory and coordinating control incorporated, but lacks the system-wide control rules that depend on the downstream WRRF, and therefore not all events can be equally well simulated. The provided rain data give an indication of the spatial variability of rainfall, especially for the more extreme events. The extensive observations obtained in the area show that it is not always easy to operate sensors, but also that there is a great potential in using data to a much greater extent than previously. With the provided data set, all researchers have access to the same models and data, which can enable a boost in research and innovation in the future.

## Author contributions

All authors jointly contributed to conceptualizing and designing the study, discussing results and drafting or revising the manuscript. ABK and ANP designed the observation program. ANP collected and curated the observation data, and ANP and JWP prepared the software for data cleaning. ANP prepared the MU models, whereas AVR and JWP prepared the SWMM model. ANP, JWP, AVR and MB prepared the initial draft manuscript, and ANP, JWP, MB and PSM prepared the visualizations. ANP and PSM revised the manuscript and prepared the final submitted manuscript, which was approved by all authors. ABK, MB and PSM supervised the project.

## Competing interests

The authors declare that they have no conflict of interests.

**Acknowledgements**

The authors are grateful to several organisations agreeing to freely share data for this case area, especially VCS Denmark
which provided all the asset data, the in-sewer observation data and some of the rain data and the radar data, the Danish
Meteorological Institute (DMI) and the Water Pollution Committee (WPC) of The Society of Danish Engineers who provided
rain gauge and weather-radar data, and the Danish Agency for Data Supply and Efficiency (DADSE) who provided spatial
data as ortho photo and digital terrain models. The rain gauge data can be used freely for teaching and research, with appropriate
indication of the original source, but are according to an agreement with DMI and WPC not for commercial use. The data from
DADSE can be freely used with appropriate indication of the original source. DADSE retain copyright to the data when it is
passed forward. Details of how to cite the data are given in the data set (Pedersen et al., 2021a).

**Financial support**

The work done by Agnethe Nedergaard Pedersen was partly funded by the Innovation Fund Denmark (file no. 8118-00018B),
the work of Antonio Vigueras-Rodriguez was funded by a research stay programme of the Spanish Ministry of Science,
Innovation and Universities (Ref.: PRX19/00230), and the work of Jonas Wied Pedersen and Morten Borup was partly
supported by the European Regional Development Fund through the NOAH Project (Interreg Baltic Sea Region Programme
Grant #R093).

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
