# Peer review of "The Bellinge data set: open data and models for community-wide urban drainage systems research"

_Earth System Science Data, 2021_

## Author Comment (AC2)

Manuscript ID: essd-2021-8: **The Bellinge data set: open data and models for community-wide urban drainage systems research,** by Agnethe Nedergaard Pedersen, Jonas Wied Pedersen, Antonio Vigueras-Rodriguez, Annette Brink-Kjær, Morten Borup and Peter Steen Mikkelsen

**Detailed point-to-point responses to comments from reviewers**

|  | Reviewer comment | Author response
*References made to locations in the manuscript within this column refer to the revised manuscript* |
|---|---|---|
| **RC1 – Anonymous referee #1 ([https://doi.org/10.5194/essd-2021-8-RC1](https://doi.org/10.5194/essd-2021-8-RC1) )** | | |
| General | | |
| RC1.0 | It is quite necessary and important to develop an open data and models of urban drainage systems for both academia and industry. We indeed need a data set with good quality, close to reality for modelling, analyzing, comparing performance in different control and optimization methods, etc. This work solves the problem very well.

The data and models cover a real area in Denmark, consists of up to 10-year observations for important measurements in need. Models are presented in both MU and SWMM, the most popular simulation platform, which add usability and accessibility for the potential users.

The manuscript and files in terms of data and models are organized in a good manner. Just very few comments about some confusing places: | Thank you very much for your review of the manuscript. We really appreciate this and have done our best to accommodate your suggestions for changes in the manuscript. See our replies to your detailed comments below. |
| Manuscript | | |
| RC1.1 | Line 215: "A 1 min temporal resolution applies to…" maybe better in "A 1-minute temporal resolution is applied to…"; "2 min" maybe better in "2-minute" | Thank you. We will clarify this.

The manuscript is changed to
Line 233: *"A 1-minute temporal resolution is applied to all data…"* and *"but the resolution is 2 minutes for the mobile sensors"* |
| RC1.2 | Line 365: "10 m" maybe better in "10-meter" | Thank you. This is changed in the new submission.

Line 528: *"…shorter than 10 meter is adjusted to 10 meter for…"* |
| **Assetdata.pdf** | | |
| RC1.3 | It is nice that some important variables have been translated into English, which indeed increase its readability.

In the row of **NetworkCat**, there appeared "?", what does that mean? Is it possible to put the real translation here?

In the row of **StatusCode**, the 7, 8, 50 of K_STATUS failed to be translated, will that be a problem for the readers? | Thank you for pointing this out. The Table is from the DanDas manual and contains some entries that are not relevant for the Bellinge data set. We have focused on adding English translations only of terms relevant to the Bellinge data set, but realise now that a few more needed translation. We have thus replaced "?" (which was an error) with *"Interceptor pipe"* and added translations of StatusCode 7, 8, and 50 *("Constructed", "Removed"* and *"Other")*. |
| **G71F04R_Level1_System2000p2_proc_v6.xls** | | |
| RC1.4 | "ffill" means "fill", is it necessary to modify? | The data supplier that VCS uses for many of its in-sewer sensors reduces storage requirements of |

| | | measured values by leaving empty data points in the time series if consecutive measured values are (near) constant. Consecutive points are seen as constant if water levels vary less than one centimeter (manual setting) between two consecutive time steps. So, "ffill" means "forward-fill" as empty data points are filled with the value of the most recent data point.

 The following changes is suggested in the manuscript.
 Line 321ff: "*Data from iFix is set to reduce data storage requirements at the sensor by leaving out observations that changes in the coming timestep below a given threshold (for water level sensors most often set to 1 cm). Therefore, this should not be seen as a period of failure, or signal loss, and for this script forward-filling is applied to the values which have these properties.*" |
|---|---|---|
| **CatchmentDescription.pdf** | | |
| RC1.5 | is it possible to give an explanation about meaning of standard and Other, how to define them, is it possible to provide more information about them? | The "SA (standard)" and "Other" categories refer to two different sets of imperviousness parameter values characterizing different land-use classes. The standard is what is far most used in the service area of the utility company and have emerged based on experience. The "Other" category is an adjustment of the imperviousness related to the catchment area of G71F68Y based on interpretation of observations many years ago. This is now better explained in the document. |

---

## Author Comment (AC3)

Manuscript ID: essd-2021-8: **The Bellinge data set: open data and models for community-wide urban drainage systems research,** by Agnethe Nedergaard Pedersen, Jonas Wied Pedersen, Antonio Vigueras-Rodriguez, Annette Brink-Kjær, Morten Borup and Peter Steen Mikkelsen

**Detailed point-to-point responses to comments from reviewers**

|  | Reviewer comment | Author response *References made to locations in the manuscript within this column refer to the revised manuscript* |
|---|---|---|
| **CC1 – Community referee (https://doi.org/10.5194/essd-2021-8-CC1)** | | |
| CC1.0 | Dear Authors, Thank you for sharing such a comprehensive dataset with the community. In my opinion, your paper is a very important contribution, since it fills a gap in the field of urban hydrology, which lacks open data. While having a look at the SWMM model, I encountered an error related to the rainfall input. I think the link to the time series file is broken. In principle, this issue can be fixed easily. However, in case you plan to update the online repository, I would recommend to update the inp file of the model in order to make it easier for people who just started modelling with SWMM. Anyway, I really like your study and I am looking forward to your final revised paper! Best wishes, Kristian Förster | Dear Kristian Förster, Thank you very much for your feedback and comment. We have modified the file in the repository with the suggested change and hope that this change makes it easier for users to apply. We look forward to hear about your future work based on the data set. Best wishes, The authors. |

---

## Author Comment (AC4)

Manuscript ID: essd-2021-8: **The Bellinge data set: open data and models for community-wide urban drainage systems research,** by Agnethe Nedergaard Pedersen, Jonas Wied Pedersen, Antonio Vigueras-Rodriguez, Annette Brink-Kjær, Morten Borup and Peter Steen Mikkelsen

**Detailed point-to-point responses to comments from reviewers**

|  | Reviewer comment | Author response
*References made to locations in the manuscript within this column refer to the revised manuscript* |
|---|---|---|
| **RC2 – Anonymous referee #2  (https://doi.org/10.5194/essd-2021-8-RC2)** | | |
| **Summary** | | |
| RC2.0 | The submission is a timely contribution in the field of urban drainage, making use of the emerging possibility to publish articles on original research data. Presented data (and models) are expected to enable researchers to re-evaluate difficult-to-obtain data in context of urban drainage modelling, to study the influence of different sources of precipitation on hydraulic simulations, and to apply different data analysis techniques, e.g. to detect anomalous sensor recordings.

The manuscript is well structured and well written. A short review explains the underlying motivation for sharing data and models; reference is given to previously published data publications. The main part describes diverse data sets and corresponding hydraulic models related to a small Danish combined sewer network. The study area and its urban drainage system are very well described. The authors provide supplementary information on sewer infrastructure and hydraulics (models; photo documentation; very illustrative drawings on structures; geodata) allowing for an adequate interpretation of the complex drainage situation. Individual data sets are explained in the text and in a separate document accompanying the data package.

While it is clearly acknowledged that collecting and compiling this data set has been a great effort, and a publication of topic and data is principally recommended, I see the following key points that need to be addressed in a revised version: | Thank you very much for your thorough and constructive review of the manuscript. It is really appreciated, and we thrilled to discuss with you and the urban drainage community in connection with this review process, and hopefully in future works aimed at more open data in urban drainage research. We have done our best to accommodate your suggestions, see our replies to your detailed comments below. |
| **Data ownership** | | |
| RC2.1 | Data ownership: I am wondering if data, that is publicly available anyway (meteo data, topographic data) should be - at least - specifically labelled in order to allow a differentiation from data collected on purpose, such as in-sewer observations. It should be discussed (also in the community) how this should be handled, i.e. it needs clear | Good discussion point. Our aim was to provide as much data that we knew was available for easy access to the user, with clear explanation of who owns the data. This was done by addressing the ownership with references in the paper, and in the repository. As this seems not to be sufficient, we have now included a new figure in section 7, Figure 13, which highlights the ownership of the data and also indicates what data types can under normal |

| | | |
|---|---|---|
| | statements to clarify data ownership of original data. | circumstances be freely accessed and which data cannot (private data, third-party open access data, third-party subscription data, open data). See also our reply to comment RC2.5). |
| **Hydraulic models** | | |
| RC2.2 | Hydraulic models: The authors attribute large parts of the manuscript (7 of 24 pages) to the comparison of two hydraulic model implementations that describe the same case study system (one being a modified export of the original). While comparing the effect of different conceptual approaches for surface runoff models may generally be an interesting aspect, the key focus (which is the data set) is - in my opinion - unnecessarily diluted by elaborating upon structural model uncertainty. I suggest streamlining the study here. This could be accomplished either by focusing on one model implementation only, by outsourcing the model comparison, and/or by discussing the models usefulness, e.g. to check the plausibility of observations. | Thank you for this comment. Model comparison is not the main focus of the paper, but the provided models represent important system knowledge and are thus important. However, our rationale for providing two model implementations (MU and SWMM) was perhaps not clear in the original manuscript, and we have thus added the following text to the revised manuscript (Lines 440-443): *"VCS uses MU in the daily modelling and model updating work, which is however not easily accessed by potential users of the data set because of its proprietary nature. We therefore also provide a SWMM model (created to mimic the behaviour of the MU model), which is open source and thus readily available for use by the international research community."*

Acknowledging that MU and SWMM have many similarities but also some differences that may yield different model results, we find a brief comparison of model-to-model as well as model-to observations (Figure 11) appropriate. |
| | | |
| RC2.3 | Only vague information is given on what to do with the presented data and models. The very last paragraph provides a glimpse and mention the "great potential in using data to a much greater extent than previously. Provide more concrete examples, i.e. ideas how to utilise the data. This should support/illustrate the value, uniqueness and usefulness of this research data publication.

Please find more elaborated comments, split in major and minor aspects, below. | Thank you for the comment. We do in the submitted manuscript provide our view on potential research in the Introduction (Line 55-63), but we acknowledge that this may not be sufficient. We have therefore implemented the following changes to the manuscript to clarify our intentions.

A new section is inserted, "6 Potential use of the data set" (Lines 629-654). The original manuscript had (by mistake) no section 6, so this new section was easily inserted without requiring other structural changes to the paper. We have moved the section mentioned above from the Introduction to this new section, and deepened it substantially to highlight what we know the data set is currently being used for and may potentially be used for by the international urban drainage community in the future. Instead, we have added the following sentence in the Introduction (Lines 80-84): *"The selection of data and models provided here aims to be as "open-minded" as possible, and we believe it can be used to initiate research across a* |

|  |  | *range of highly relevant topics, and also inspire discussions among water utilities on the benefits of high-quality data acquisition and modelling".* Finally, we now refer to Sect, 6, 7 and 8 in the last paragraph of the Introduction (lines 88-89). |
|---|---|---|
| **Major points:** | | |
| Handling the data and meta-data | | |
| RC2.4 | Data is provided in nine (9) individual packages (ZIPs) through a university hosted research data repository. Downloading, sorting, and renaming inconsistently named data packages takes a while. This should be organised in a more stringent manner, the file naming should revised and occasional redundancies be eliminated. | Thank you for that insight. Our intention was that the entire folder is downloaded, but we do acknowledge that when downloading the individual files, the naming may be difficult to cope with. The reason for the split in several folders is that we expect not all readers to be interested in the entire dataset.

To facilitate easy navigation of the data folders and files in the repository, we have now incorporated Figure 13 in the manuscript and updated some of the folders with new names referring to the number of the subfolder. In addition, we have added the following explanation (Lines 680-684): *"The data set is split into 9 items as there is no need to download all for a very specific use. Figure 13 gives an overview of the data repository with clear identification of both the ownership of the data and how the data would normally be accessed, prior to publishing this data article and repository. The provided data comes with a Creative Commons license CC BY 4.0, except from some of the rain data from DMI which comes with a CC BY NC 4.0 license (commercial use not permitted)."* |
| Ownership of the data | | |
| RC2.5 | The data itself is actually a compilation of various data sets, of which some are acquired in own or contracted field measurement campaigns (water level, flow sensors, i.e. sensor data - #2; CCTV data - #5), some data stem from publicly available sources (radar data: VCS Denmark; Orthophoto, Digital Terrain Model: SDFE), or from sources where data typically need to be purchased (rain gauge data - Danish Meteorological Institute). One question I would like to put up for discussion here: is it scientifically innovative to publish compilations of different data sets that are, on the one hand, available (anyway) and, on the other hand, selectively undercut with own or specifically contracted field measurements? | Thank you for the comment. We are not sure what is exactly meant by "undercut" but provide and answer to the best of our ability.

We agree that it is not scientifically innovative to publish available information only. However, this paper presents a data set gathering multiple data sources that are not all easily accessible for researchers. We have added the following additional explanations in the new section "6 Potential use of the data" (Lines 629-654), see also our reply to comments RC2.3): *"The water sector is furthermore known for inadvertently "hiding" data in silos hosted both within utilities (e.g. in different departmental systems) and by different external contractors, which makes integrated analysis tedious and resource demanding (e.g. Lund et al., 2021). We thus also encourage discussion on how the various information sources provided here may work* |

| | | |
|---|---|---|
| | | *together as required in future digital twin environments (Pedersen et al., 2021b)."*

The dataset is curated based on our perception of what a potential data usage can be, with an ambition to not be narrow-sighted. |
| **Usability of data and models** | | |
| RC2.6 | There is a very short and unspecific section on what to do with this data in the beginning (line 81 ff.) and a more concrete paragraph in the conclusion (493 ff). The latter section would deserve a more in-depth elaboration in a previous chapter. More concrete examples should be provided to illustrate the value, uniqueness and usefulness of this data publication. For instance, what is the added value of providing CCTV inspection data? | Thank you for your comment. We refer to our answer to comment RC2.3, where we explain that a new section "6 Potential use of the data set" has been inserted (lines 629-654).

Besides that we see a great value of having CCTV included in the data set, as this provides valuable information of understanding the pipes, the manholes and the constructions which is not always easy to obtain from top-down pictures. This is particularly addressed in lines 636-368. |
| **Missing meta-data on sensor readings** | | |
| RC2.7 | In line 250 ff it is stated that "Exact documentation of sensor maintenance has not been a high priority over all the years, and it is therefore presently not possible to give an overview of when and where sensors have been repaired, replaced or received some sort of maintenance." That is, meta-data or log files are not provided. Comments such as, "The 0-point may have changed during the years, and there is no log-file with changes in SCADA settings in System2000…." (line below the Fig. 1 caption in Sensordata.pdf) are honest, but not very helpful. This is a drawback, which clearly limits the possibilities to interpret in-sewer sensor data. In a didactical example, the authors indeed provide two cases, which illustrate how this can effect sensor data interpretation. But, how can data from other sensors be interpret if I do not know zero-point has changed? | Thank you for your comment. We note that we here provide more than 10 years of observation data from a utility company that has started data collection before many others, and that critical issues with the presented data set can provide learnings that others may benefit from. By exploiting the data we can maybe become better at logging the information that is needed in the future. We acknowledge that the logging of for instance the zero-point could have been useful, but maintain that the data still consist value for example as a basis for developing anomaly detection and data reconstruction software.

We have implemented the following changes to the manuscript.

Line 295-297: we added: *"We acknowledge that this is not optimal. The utility company is in a transition process of changing the way meta data is logged. By exploiting the procedures in a typical Danish utility company we can hopefully start a discussion of how to make best practices."*

in the new section "6 Potential use of the data" we have added (lines 642-645): *"With increased focus on digitalisation, the data set can also be used to initiate discussions on data acquisition and transfer needs (Eggimann et al., 2017) in order to gain insight into urban drainage systems that are gradually becoming more complex (Blumensaat et al., 2019), and to initiate discussion about what metadata and* |

| | | |
|---|---|---|
| | | *logs should be stored to ensure available information for future use."* |
| **SWMM and MU model** | | |
| RC2.8 | A large part of the manuscript (7 of 22 pages) actually focuses on model-related issues, i.e. it discusses effects of conceptual differences in two urban drainage models. Since most data users would use the SWMM model for simulations, my comments mainly relate to the SWMM model implementation. While my general comment on the model comparison in the summary section remains the major point of critique, the following model-related aspects appear odd and need some clarification: | See our reply to comment RC2.2. |
| RC2.8.1 | It is not clear why two different versions of the MU model are provided. If the "Mike Urban model of the system anno 2020" represents the system "as it looked medio 2020, but it is a good representation of the system from 2010 and onwards.", and no significant land use changes were observed/assumed (cf. line 73 f.) it is not clear what the user should do with the old SWMM model implementation. In order to avoid ambiguities I suggest excluding irrelevant data and model files, such as the old model version. If still relevant, please explain why. | The model of the system before 2009 can make it clearer to some readers why the current solution (underground storage-pipe and basin, to avoid overflows) was implemented, and also provide information to interested readers about why some pipes are very large in the system due to the objective of the old system. Therefore we still think that the old MU model is relevant to share. We have modified a sentence in section 2.3.2 so that it now reads (line 192-194): *"Models are available for both the old system (2009, may be useful when seeking to understand the historical evolution of the system) and for the new system (from 2010, may be useful when comparing with observation data) and can be found in the data set (Pedersen et al., 2021a)."* |
| RC2.8.2 | In the model description document (pdf) it is mentioned that "the [SWMM model] parameters for the infiltration is currently set extremely high so that infiltration from green areas will not appear". This is most likely a typo, since the sentence does not make sense in this context. Please correct the typo. NB. Generally it can be stated that tweaking the parameters of the SWMM implementation in such a way that runoff-efficient areas are reduced to only impervious areas can be critical when having an average degree of imperviousness of 35 % (as it is the case here). It could further be discussed how this effects simulation results. | Agreed, this is a typo, we thank the reviewer for pointing this out. We have replaced the word "infiltration" with the word "runoff" in the model description document. With regard to the difficulties of modelling rainfall-runoff from not paved-areas, we have now emphasized this as a potential future use of the data set in the new section "6 Potential use of the data set" (see also our reply to comment RC2.3), |
| RC2.8.3 | In terms of plausibility of the SWMM model performance: a moderate rain event of about 13 mm h-1 leads to flooding of several nodes in the network (event early of 29-Jun-2012). Either the system is poorly designed, or the model is hydraulically incorrect. The potential overestimation of the flooding activity may | 13 mm h-1 corresponds to 3.6 um s-1 over an hour, which is indeed represented by the event 29-June-2021 and in Bellinge statistically occurs with a return period just below 0.5 years, cf. Figure 10. However, the 10 minute intensity of this event statistically corresponds to a return period of 2 years, and the 5 minute |

| | | |
|---|---|---|
| | also be discussed in the context of other peculiarities identified. | intensity corresponds to a 5 year return period. In both MU and SWMM we experience app. 110 nodes of 1022 (~11%) that are flooded, which is quite normal for an elderly combined system.

However, we do recognize the need for better tools to investigate the performance of the models. This will be discussed in future papers using the data set. |
| RC2.8.4 | Sewer infiltration is completely neglected, despite the fact an internal report says that it makes up 30 % of the hydraulic loading at the catchment outlet. This is a significant share, which IMO cannot be neglected when considering the model performance (dry and wet weather). Can you provide more information from the internal report on how the 30 % infiltration is quantified? Could you describe unsuccessful attempts to implement infiltration in the models? This could be useful for data users when trying to find alternative solutions. | Thank you for your comment. Yes, 30% infiltration-inflow is a significant share. VCS recognizes that and has struggled to find a good way to incorporate this. The 30% results as a rough estimate from analyzing pumping station data, and VCS has earlier (via a consultant) looked into machine-learning techniques applied to observations near the treatment plant (which cannot be directly applied to upstream areas). These investigations have not been documented in detail, and we have not attempted to qualify this further in the context of preparing this open data set. This could, however, be investigated further based on the data set and models provided (now mentioned in the new section "6 Potential use of the data set"), cf. our reply to comment RC2.3. The following changes are suggested to accommodate your questions.

Line 498: We have added *", based on analysing data from pumping stations, "*

Line 501ff now reads*: "Several attempts have been made in VCS to model the infiltration-inflow, for example machine-learning techniques applied to observations near the treatment plant. These can be used for estimation of the inflow to the treatment plant but seldom matches reality when scaled to upstream catchments. Therefore, infiltration-inflow was not included in the MU model provided here but we encourage potential users of the data set to investigate this further".* |
| RC2.8.5 | SWMM Infiltration parameters are tweaked to the extreme to match observations. At line 330 ff, the authors however state "VCS has a philosophy of transparency in models, where understanding the system behaviour is more important than ensuring a perfect calibration with non-transparent parameter sets, meaning that VCS does not want to tune conceptual parameters to unrealistic values | The SWMM surface runoff module has been tweaked to simulate the behaviour of the Mike Urban surface runoff module. The utility company VCS uses Mike Urban in its daily operation, and this model has not been calibrated with parameter estimation. We suggest the following changes to clarify this in the manuscript. |

| | | |
|---|---|---|
| | in order to fit models to observations.". While this is sustainable opinion, it is somewhat contradictive to the parameter tweaking. Please clarify! | The sentence in line 359-361 of the original manuscript is changed/expanded and now reads (Lines 521-524): *"In order to make the two models as similar as possible, the parameters for pervious areas in SWMM's infiltration module were thus set to unrealistically high values, so that rainfall on such surfaces readily infiltrates into the ground instead of producing runoff to the urban drainage system. On impervious surfaces, the parameters were set to produce run-off similar to the runoff simulated with the MU model"*. |
| **Minor points** | | |
| RC2.9 | Data validation - Chapter 3.4: the section on data cleaning could be more elaborative; reference should be given to existing works, e.g. (Leigh et al. 2019). Five different methods are explained for data validation (cleaning), whereas two of them are subjective ("manual remove"; outlier detection for interim Danova sensor). Furthermore, it remains unclear why only gaps shorter than 5 minutes have been interpolated, why not up to 10, 15 minutes? Why would it be necessary to interpolate them at all? | Thank you for the comment. This reference is very interesting and we have thus added it to the sentence appearing on Line 311ff: "*leave a more comprehensive data validation as a research opportunity for ourselves or others in the future e.g., (Leigh et al., 2019)*"

Besides that the following sentences is slightly changed:

Line 308-310: "*For this release, it was for practical reasons, decided to use an initial set of common, simple data cleaning techniques and leave a more comprehensive data validation as a research opportunity for ourselves or others in the future e.g., (Leigh et al., 2019).*"

And line 326-328): "*Simple gapfilling based on linear interpolation was done for gaps shorter than 5 minutes, as increased gap-filling period would increase the risk of interpolate a potential peak.*"

We recognize that manual remove can be subjective and depends on the purpose of the calculation, whereas outlier detection seek to identify problematic spike values that appears constantly during a timeseries. We acknowledge that potential real outliers may be removed with the technique, but also emphasize that this is not the focus of this manuscript. |
| RC2.10 | Meteorological variables from DMI should be referred to in Ch. 3 since these can also be considered as "observations". Showing the 10 year time series in Fig. 2 illustrates the availability but has no added value. | We understand the logic of you comment. However, Figure 2 is placed in the chapter with system description (Ch. 2), because meteorological information is key to understanding the climatology and temporal variability of hydro-meteorological system in Bellinge, just as the maps with topographic elevation curves in Figure 1 is important to understand the water infrastructure. To make |

| | | |
|---|---|---|
| | | this more clear, we have changed the header of section 2.2 (from "Meteorological variables" to "Climate and meteorology of the area") and modified the first sentence that now reads (line 137): *"The area has a warm, temperate climate. Several meteorological variable are available …"*. |
| RC2.11 | Figure 7: it is not clear what type of sensors the GF73F010 and GF72F040 are. Please specify in the Y-axis or caption. | A change in the caption is suggested to: "*G73F010 and G72F040 are level sensors.*" |
| RC2.12 | Figure 8: inconsistent caption formatting. | The caption is entirely bold now. |
| RC2.13 | Sentences like the one in line 330 ff. are rather opinions than solid research results. Please consider rewriting these sentences (without changing the valuable meaning) and provide references, if possible. | Calibration of urban drainage models has been subject to a lot of research internationally for more than a decade, with little apparent progress in practice. We do however not wish to enter into a detailed discussion about the calibration topic, which is important but not key to the present paper. This is why we have simply stated the rationale that VCS relies on when maintaining their hydraulic models. We have however now modified the paragraph slightly and added a number of references, so that it now begins with (Line 554ff): *"Calibration of urban drainage models has been subject to a lot of research internationally for more than a decade (e.g. Bach et al. (2014), Broekhuizen et al. (2020), Nagel et al. (2020), Tscheikner-Gratl et al. (2016) and Vezzaro et al. (2013)). However, VCS has a philosophy of transparency in models…"* |
| RC2.14 | Descriptions for some data sets are very sparse, for others they are sufficiently comprehensive. Some of the accompanying documents are somewhat sloppily prepared and need revision (e.g. Models.pdf). | We have revised all the documents accompanying the data. |

**References:**

Bach, P.M., Rauch, W., Mikkelsen, P.S., Mccarthy, D.T., Deletic, A., 2014. A critical review of integrated urban water modelling - Urban drainage and beyond. Environ. Model. Softw. 54, 88–107. https://doi.org/10.1016/j.envsoft.2013.12.018

Broekhuizen, I., Leonhardt, G., Marsalek, J., Viklander, M., 2020. Event selection and two-stage approach for calibrating models of green urban drainage systems. Hydrol. Earth Syst. Sci. 24, 869–885. https://doi.org/10.5194/hess-24-869-2020

Leigh, C., Alsibai, O., Hyndman, R.J., Kandanaarachchi, S., King, O.C., McGree, J.M., Neelamraju, C., Strauss, J., Talagala, P.D., Turner, R.D.R., Mengersen, K., Peterson, E.E., 2019. A framework for automated anomaly detection in high frequency water-quality data from in situ sensors. Sci. Total Environ. 664, 885–898. https://doi.org/10.1016/j.scitotenv.2019.02.085

Lund, N.S.V., Kirstein, J.K., Madsen, H., Mark, O., Mikkelsen, P.S., Borup, M., 2021. Feasibility of using smart

meter water consumption data and in-sewer flow observations for sewer system analysis: a case study. J. Hydroinformatics 795–812. https://doi.org/10.2166/hydro.2021.166

Nagel, J.B., Rieckermann, J., Sudret, B., 2020. Principal component analysis and sparse polynomial chaos expansions for global sensitivity analysis and model calibration: Application to urban drainage simulation. Reliab. Eng. Syst. Saf. 195. https://doi.org/10.1016/j.ress.2019.106737

Tscheikner-Gratl, F., Zeisl, P., Kinzel, C., Rauch, W., Kleidorfer, M., Leimgruber, J., Ertl, T., 2016. Lost in calibration: Why people still do not calibrate their models, and why they still should - A case study from urban drainage modelling. Water Sci. Technol. 74, 2337–2348. https://doi.org/10.2166/wst.2016.395

Vezzaro, L., Mikkelsen, P.S., Deletic, A., McCarthy, D., 2013. Urban drainage models - Simplifying uncertainty analysis for practitioners. Water Sci. Technol. 68, 2136–2143. https://doi.org/10.2166/wst.2013.460